# Pathogenicity and transmission of *Morganella morganii* in honey bees

Yijun Chen, Qiang Huang *

Honeybee Research Institute, Jiangxi Agricultural University, Nanchang, China

* qiang-huang@live.com

## Abstract

Honey bees provide essential pollination services in the ecosystem. The high annual loss of honey bees has raised concerns about global food security and the agricultural economy. As a primary stressor causing colony failure, the mite *Varroa destructor* feeds on the hemolymph and the bee's fat body tissue. The *Varroa* mite-associated deformed wing virus has been extensively studied because it can be found in each individual mite and causes bee mortality. A recent study shows that the *Varroa* mite can transmit pathogenic bacteria, while the transmission route remains unclear. In this study, we isolated and assembled a previously uncultured bacterium, *Morganella morganii*, from the mites *Varroa destructor*. This pathogenic bacterium exhibited a high case fatality rate, as evidenced by 215 cells causing over 30% mortality in pupae and adult bees. Using a fluorescent protein-tagged strain, we provide evidence that *M. morganii* can not be transmitted among bees through social contacts, while it can be transmitted from mites to bees, and vice versa. The cumulative incidence of transmitting *M. morganii* from infected bees to mites is 92.1%, and 68.49% from infected mites to naïve bees. Our data aligns with the honey bee colony collapse in winter, when the mite population expands, accelerating the honey bees to tap into a reservoir of this lethal bacterium.

### Author summary

Honey bees are the most abundant commercial pollinator, and beekeepers need to treat *Varroa* mites twice a year to prevent colony collapse. The mites suck the hemolymph and the bee's fat body tissue, and vector several viruses. However, the *Varroa* mite-associated bacteria have been rarely investigated. In this study, we isolated a lethal bacterium, *Morganella morganii*. A low dosage of *M. morganii*, which is transmitted through mite feeding, causes high mortality in pupae and adult bees. Our data deepens the knowledge about the bacterial pathogen, vectored by mites.

**Data availability statement:** All study data are included in the article and supplementary material. The genome and gene annotation are deposited in NCBI GCA_044772875.1. The data used to generate the figures in the main text and supplementary material are provided in S1 Data.

**Funding:** This study is funded by the Jiangxi Double Thousand Plan (#jxsq2020101078 to QH). The funder has no roles in the study design, data collection and analysis, decision to publish, or preparation of the manuscript.

**Competing interests:** The authors have declared that no competing interests exist.

## Introduction

The honey bee, *Apis mellifera,* is a crucial domesticated pollinator that substantially improves fruit set for flowering plants. Recently, the annual loss of honey bee colonies has approached 60% in North America [1], and the mite *Varroa destructor* is the primary stressor causing colony failure [2–4]. *V. destructor* is an ectoparasitic mite that feeds on the hemolymph and bee's fat body tissue [5]. The *Varroa* mites parasitize larvae, pupae, and adult bees [6]. Once parasitized by the *Varroa* mites, honey bees exhibit immature aging [7], impaired flying and cognition [8,9], and shortened lifespan [10,11]. Upon feeding, the mites can inject a variety of viruses into the hemolymph of bees: among them, the deformed wing virus (DWV) is one of the most prevalent, causing developmental disorders in infected individuals [12,13]. The naïve *Varroa* mites can also acquire the viruses when feeding on infected bees [14].

DWV has been extensively investigated because of its ubiquity in *V. destructor* [12]. DWV can be horizontally transmitted from bee to bee, as validated by fluorescent tagging, and infected individuals serve as a reservoir for naive mites to acquire and subsequently transmit the virus [14]. The *Varroa* mites also harbor several bacteria: a previous 16S sequencing study identified a few bacteria in mites and honey bees, including *Morganella*, *Spiroplasma*, and *Enterococcus*, suggesting these bacteria may transmit between them [15,16]. Given the difficulties in cultivating these pathogens and the complexity of the bacterial community found in *Varroa* mites, this field of study remains largely unexplored. Until recently, *Enterococcus faecalis* was isolated from dead bees, which is vectored by *Varroa* mites. *E. faecalis* caused substantial mortality in the honey bee *A. mellifera* [17]. Thus, the mite also serves as a vector for pathogenic bacteria of honey bees. However, the transmission routes of *E. faecalis* are not fully understood.

In a previous study, a few bacterial pathogens were inferred in both mites and honey bees [16]. We hypothesize that the *Varroa* mite transfers and acquires bacterial pathogens upon feeding the bees. To investigate the virulence and transmission routes of the bacteria vectored by *V. destructor*, we first isolated and cultured those bacteria. Then, we injected the bacterial inoculum into bees to observe the bacterial proliferation and bee mortality. Specifically, we isolated the first strain of *Morganella morganii* in *Varroa* mites, which is highly virulent in honey bees. Unlike the DWV, *M. morganii* can't be transmitted from bees to bees. Instead, transmission of *M. morganii* depends on *Varroa* mites. Honey bees and *Varroa* mites both serve as a reservoir of *M. morganii*, facilitating the mites in acquiring and transmitting it. Our data revealed a high lethality induced by *M. morganii* through mite bite: this might partially explain bee colony losses during late autumn and winter, when intraspecific contact among bees is high.

## Materials and methods

### *Morganella morganii* isolation, morphology, and growth

We collected *V. destructor* mites from adult honey bees *A. mellifera*, using the sugar-shaking method from multiple colonies in the experimental apiary in Jiangxi Agricultural University [18]. We collected and rinsed 50 mites under distilled water,

pooled each of 5 mites, and homogenized them in 500 μL Phosphate Buffered Saline (PBS). We further diluted the homogenates to $10^{-3}$ and plated 50 μL on Lysogeny Broth (LB), Columbia Blood Agar (CBA), Trypticase soy agar (TSA), Potato Dextrose Agar (PDA), Brain Heart Infusion Agar (BHI). We prepared two plates for each culturing medium and incubated one plate in a $CO_2$ incubator and the other in a regular incubator. Single colonies were collected and used for 16S sequencing to infer the bacterial species. The morphology of single cells was photographed using scanning electron microscopy. To investigate the rate of *M. morganii* proliferation, we performed a time-series recording of $OD_{600}$ in vitro. We recorded the Optical Density (OD) at 2-hour intervals until 36 hours post-plating to observe the growth rate in LB liquid medium (N = 12 replicate samples for each concentration).

## *M. morganii* genome assembly and gene annotation

The DNA of a single colony of *M. morganii* was extracted using Sodium Dodecyl Sulfate (SDS) and sheared to 30 Kbp to be sequenced in PacBio Sequel II according to the Illumina DNA preparation instructions. The highly accurate long reads (HIFI) were assembled using Flye assembler (Version 0.3-3) with default parameters [19], and the gene features were annotated using the NCBI Prokaryotic Genome Annotation Pipeline [20]. To assess the completeness of the genome, we aligned the protein sequence to the conserved single-copy orthologs of the bacterial kingdom (BUSCO Bacteria_odb10 database) [21]. The genomes' average nucleotide identity (ANI) was pairwise calculated using FastANI (version 1.34) with default parameters [22].

## Colonization of *M. morganii* and mortality in honey bees

To simulate the *Varroa* mites' feeding behavior, we injected the inoculum into the abdomen of pupae and gnotobiotic honey bees *A. mellifera*. Specifically, we prepared *M. morganii* solution at $OD_{600}$ = 1 and diluted with the PBS solution to $10^{-3}$, $10^{-5}$, and $10^{-7}$ of inoculum. To estimate the number of CFUs in each dilution, we first counted the number of cells in 1μL of $10^{-7}$ inoculum (N = 11 replicates); then, we derived the titer of the $10^{-5}$ and $10^{-3}$ dilutions, as a proportion of the average number of CFU counts in the $10^{-7}$ dilution. We injected 1μL inoculum of $10^{-3}$ (N = 127 pupae), $10^{-5}$ (N = 126 pupae), and $10^{-7}$ (N = 128 pupae) into the pupae. Additionally, we injected 1 μL PBS into 174 pupae as the control group. Dead pupae were recorded and homogenized to manually count CFU (Colony Forming Unit) using a dilution series. The remaining pupae were collected to count CFU at 7 dpi (Days Post Inoculation). Additionally, we injected 1μL inoculum of $10^{-3}$ (N = 102 gnotobiotic bees), and $10^{-7}$ (N = 116 gnotobiotic bees) into gnotobiotic adult honey bees. Again, we injected 1 μL PBS into 102 gnotobiotic bees as the control group. Dead bees were recorded and homogenized to count CFU manually. All live bees were collected to count CFU at 7 dpi.

## Oral transmission of *M. morganii*

We engineered *M. morganii* to differentiate it from the wild type that the mites might carry. The engineered *M. morganii*:pBTK520 expresses green fluorescent protein with Spectinomycin resistance. First, we investigated whether fluorescent *M. morganii* can be transmitted among bees through co-housing them. To this aim, we injected 1 μL of $10^{-7}$ inoculum into 15 gnotobiotic bees, marking them with a color patch placed on the thorax for identification. We co-housed these infected bees with 15 naive bees in a cup at 1:1 ratio with 5 replicates (N = 75 gnotobiotic infected bees and 75 naïve bees). As a control group, we injected 1 μL PBS into 15 gnotobiotic bees as the PBS group with 5 replicates (N = 75 bees). Dead bees were recorded daily and homogenized for CFU counting. At day 7, the remaining bees were homogenized for CFU counting. *M. morganii* is engineered using the Bee Microbiome Toolkit [23]. Detailed procedures are provided in S1 Appendix.

## Mite vectored transmission of *M. morganii*

We investigated the naïve *Varroa* mites' acquisition of *M. morganii* from infected bees, as well as the transmission of *M. morganii* from infected mites to naïve bees. We selected the concentration of $10^{-7}$ to investigate the transmission routes of *M. morganii* among bees and mites because this concentration allows bees to survive the experiment before being

killed by the infection. First, we injected 25 gnotobiotic bees with the fluorescent *M. morganii*, transferring one mite to each bee's body surface for 24 hours, providing 5 replicates of this experiment (N = 125 infected bees and 125 mites): we refer to this as the injection group. At 24 hours, we homogenized the mite for CFU counting to infer the proportion of mites that acquired the fluorescent *M. morganii* (N = 38 mites) from the injected bees. After 24 hours, we randomly selected 15 of those mites, transferring each one to a naïve bees' body surface for 3 days (5 replicates, N = 75 naive bees and 75 infected mites) as the recipient group. Furthermore, we injected PBS into 15 gnotobiotic bees and transferred mites to the bees' body surface at a 1:1 ratio (5 replicates, N = 75 gnotobiotic bees and 75 mites) as the control group. Dead bees were recorded daily and homogenized for CFU counting. At day 7, the remaining bees were homogenized for CFU counting.

## Statistics

The statistics were performed in R (Version 4.2.2) [24]. Colonization and persistence of *M. morganii* were analyzed using the Kruskal-Wallis test. Survival was analyzed using the Kaplan-Meier estimate in the survival package [25], and multiple comparisons were adjusted for False Discovery Rate (FDR). *M. morganii* titers within bees and mites were analyzed using the Wilcoxon rank-sum test, and multiple comparisons were adjusted for FDR. The figures were plotted using the ggplot2 package in R [26].

## Results

### Strain isolation, colonization, and genome assembly

We isolated 33 bacterial species (Table A in S1 Appendix) and successfully cultivated *M. morganii* in LB, CBA, and BHI in both regular and $CO_2$ incubators. We sequenced a single colony of *M. morganii*, assembled and annotated its genome under the NCBI genome GCA_044772875.1 (Fig 1A and 1B).

Its genome is 3.8 Mbp in a single contig, with 3,531 protein-coding genes (Table B in S1 Appendix), including 122 complete and 2 fragmented BUSCOs. We compared the isolated *M. morganii* genome with 27 previously published *M. morganii* genomes from various hosts. *M. morganii* exhibits minor genome variance between vertebrate and invertebrate hosts (Fig A and Table C in S1 Appendix). The Average Nucleotide Identity (ANI) is 92.0% with a 95% confidence interval from 91.4% to 92.6% (Table C in S1 Appendix).

### *M. morganii* induced mortality in pupae and adult bees

To investigate the impact of *M. morganii* on honey bee mortality, we injected the bacterial inoculum into the pupae and adult bees, simulating the feeding behavior of *Varroa* mites. The pupae injected with PBS showed a high survival (98.2%, 95% CI: 96.9% ~ 99.5%), while the injection of *M. morganii* resulted in a significantly high mortality at all concentrations. Specifically, the pupae survival was the lowest in the $10^{-3}$ group (34.6%, 95% CI: 25.6% ~ 46.8%, Log-rank test, adjusted $P < 0.01$) compared with the $10^{-5}$ (61.8%, 95% CI of 55.9% ~ 62.8%; Log-rank test, adjusted $P < 0.01$) and the $10^{-7}$ groups (68.2%, 95% CI of 66.2% ~ 74.8.0%; Log-rank test, adjusted $P < 0.05$, Fig 1C). The injection of *M. morganii* also resulted in low survival in adult bees. The bee survival was again the lowest in the $10^{-3}$ group (10.4%, 95% CI of 18.5% ~ 58.3%) compared with the $10^{-7}$ (60.6%, 95% CI of 52.9% ~ 69.5%; Log-rank test, adjusted $P < 0.001$) and the PBS groups (95% CI of 96.0% ~ 99.1%; Log-rank test, adjusted $P < 0.05$, Fig 1D).

To investigate whether the bee mortality was caused by the proliferation of the injected *M. morganii*, we collected dead and live individuals. We plated their homogenate of to count the CFU. We found that the CFU of *M. morganii* is $10^{13.5}$ (95% CI of $10^{13.2} \sim 10^{15.4}$) in dead pupae, while this pathogenic bacterium is rarely found in live pupae (Wilcoxon rank test $P < 0.0001$). In adult bees, *M. morganii* again showed a higher CFU in dead bees of $10^{13.7}$ (95% CI of $10^{13.3} \sim 10^{14.1}$) than in live bees of $10^{5.9}$ (95% CI of $10^{3.9} \sim 10^{7.8}$, Wilcoxon rank test $P < 0.0001$). Comparatively, the dead adult bees and dead pupae showed a similar number of CFU (Wilcoxon rank test $P = 0.35$, Fig 1E). *M. morganii* exhibited a rapid proliferation

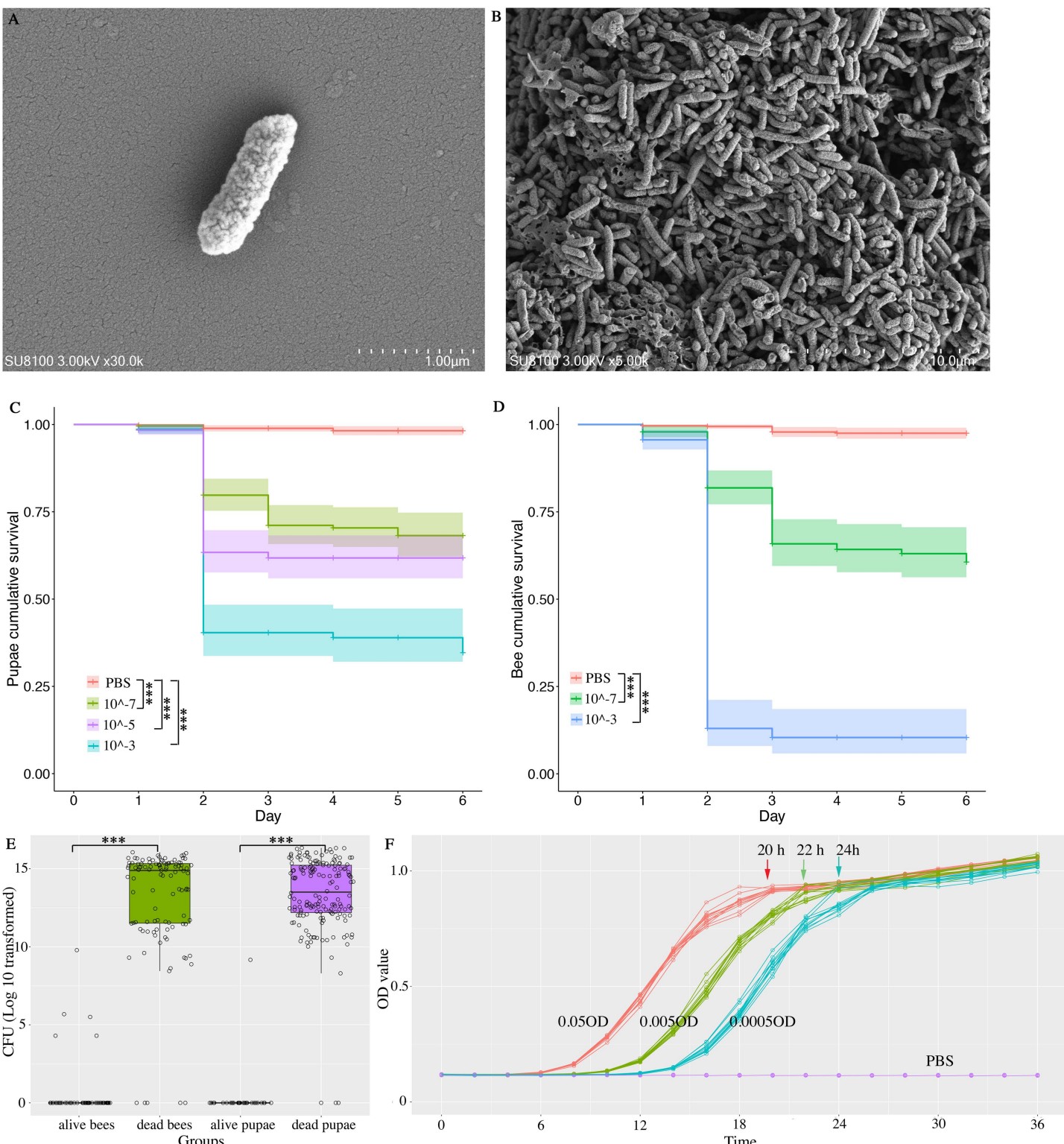

**Fig 1. *Morganella morganii* in mites.** We isolated *M. morganii* from the mite *Varroa destructor*. It appears rod-shaped under a scanning electron micro-scope (A), with no clear indication of a biofilm (B). Injection of *M. morganii* caused high mortality in pupae (C) and adult bees (D). High bacterial titer was identified in dead pupae and dead bees, which is rarely found in those that survived the *M. morganii* injection (E). The proliferation of *M. morganii* reached a stationary phase in approximately 24 hours in a series of diluted homogenates (F).

and reached the stationary phase in approximately 24 hours (Fig 1F), which may explain the observed high bacterial titer in dead bees at 2 dpi.

## The transmission of *M. morganii*

By plating the inoculum of $10^{-7}$ concentration, 1 μL of the inoculum is approximately 215 CFU (95% CI: 168~261). First, we investigated whether *M. morganii* (Fig 2A and 2B) can be transmitted from infected bees to naïve bees through co-housing. We injected *M. morganii* into gnotobiotic bees. Then, we co-housed the inoculated bees with naïve ones for 7 days. At the end of the experiment, *M. morganii* titer was high in the injection group ($10^{12.6}$ cells, 95%CI: $10^{11.9}$~$10^{13.2}$), while it was absent in the co-housed naïve bees and PBS group (Fig B in S1 Appendix). All co-housed naïve bees survived, except for one dead bee in which *M. morganii* was not detected. Thus, we conclude that *M. morganii* can't be transmitted from infected bees to naïve bees through co-housing.

Secondly, we investigated the transmission from infected bees to naïve *Varroa* mites. After introducing the mites to the PBS-injected bees, no fluorescence is observed in the mites (Fig 2C). Comparatively, after introducing the mites to the fluorescent *M. morganii* infected bees, we observed a clear fluorescence in the mites (Fig 2D). By plating the homogenates of the mites, we found the titer of *M. morganii* approached $10^{4.5}$ per mite (95% CI of $10^{4.2}$~$10^{4.9}$, Fig 3A), after parasitizing

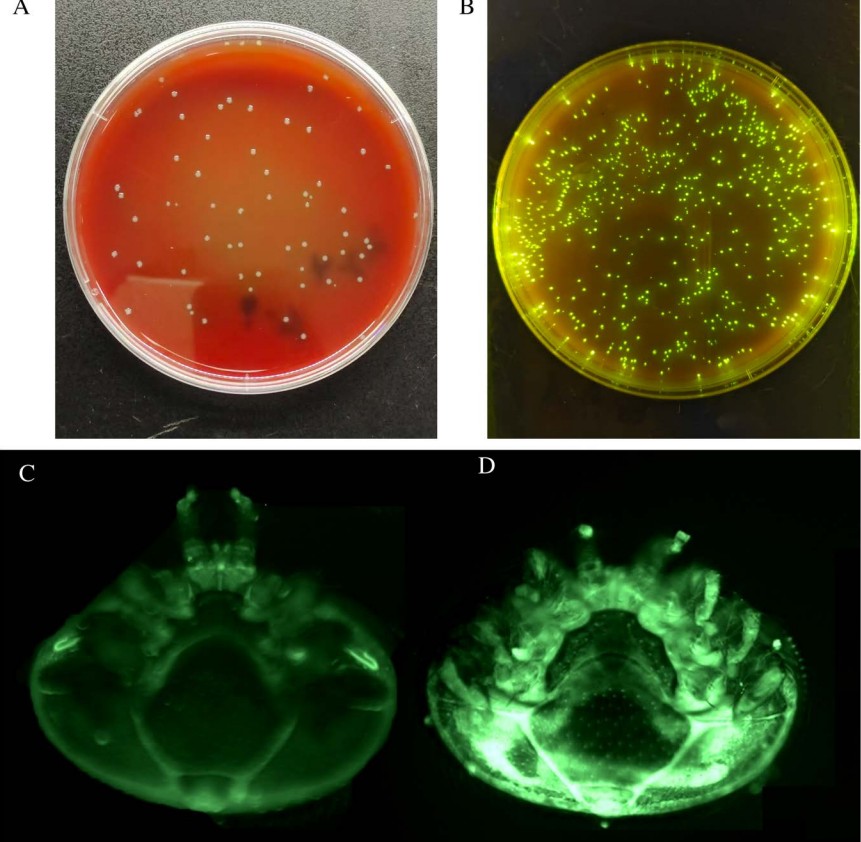

**Fig 2. *Varroa* vectored transmission of *M. morganii*.** We engineered *M. morganii* (A) to express green fluorescent protein using the Bee Microbiome Toolkit as *M. morganii*:pBTK520 (B). We inject gnotobiotic bees with *M. morganii*:pBTK520 or PBS, and then transfer a mite to the body surface of the inoculated bees. The green fluorescence is not detected in mites transferred to bees injected with PBS (C). Comparatively, the mite transferred to bees injected with *M. morganii*:pBTK520, exhibits clear green fluorescence (D).

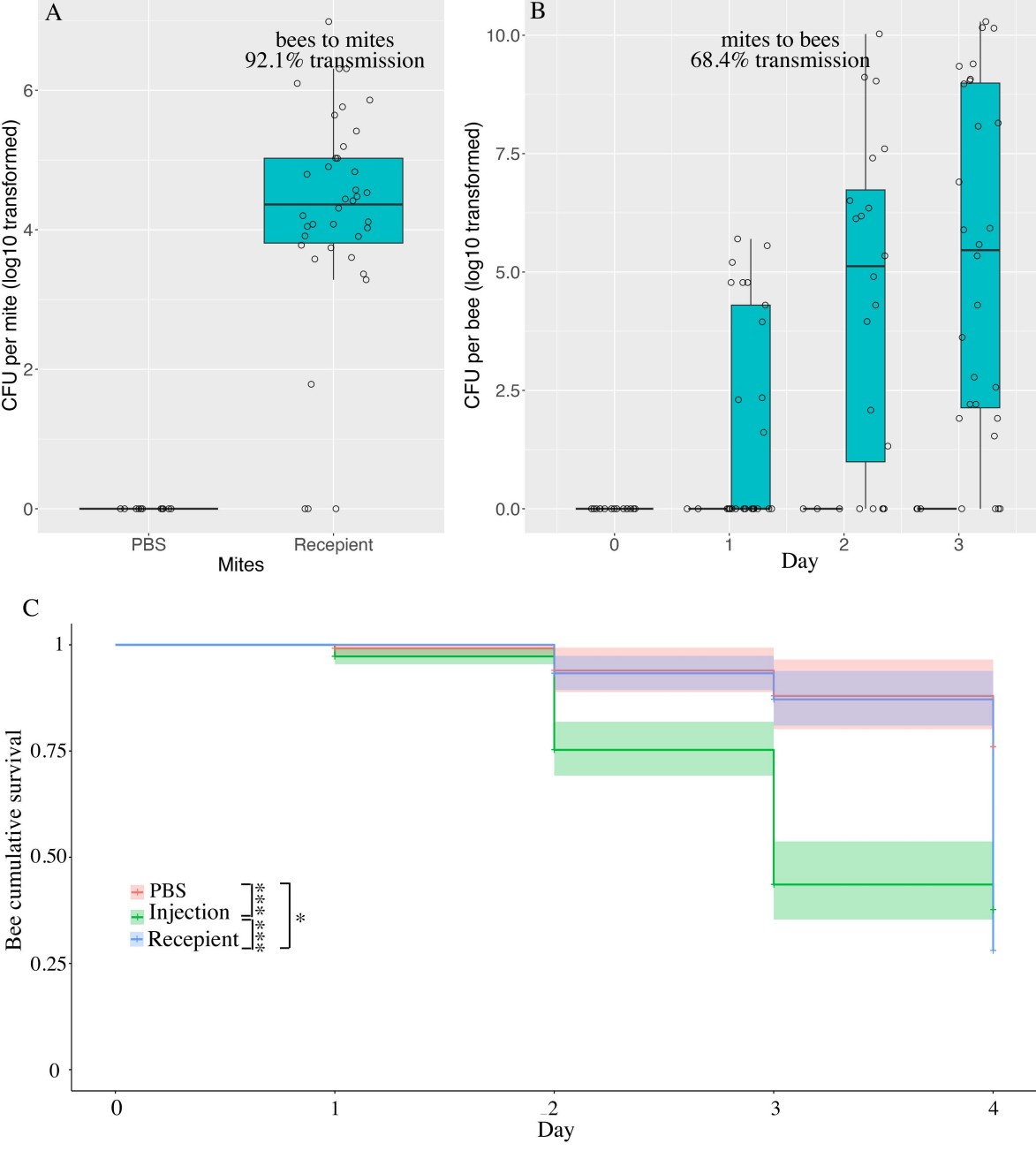

**Fig 3. Acquisition of *M. morganii* from bees and mites.** We found 35 out of 38 mites that were transferred to bees injected with *M. morganii*, harbored *M. morganii* (A). Thus, the cumulative incidence of *Varroa* mites' acquiring *M. morganii* from bees is 92.1% in 24 hours through parasitism. The mites that carry *M. morganii* vector the pathogenic bacteria to other naïve bees. The cumulative incidence of bees' acquiring *M. morganii* from mites is 68.4% (B). This pathogenic bacterium resulted in a 72% mortality rate among recipient bees, which is comparable to the 62.4% mortality rate in the injection group (C).

M. morganii injected bees, while 3 dead mites were uninfected. In total, 35 out of 38 mites harbored M. morganii once they parasitized bees that had been injected with M. morganii. Thus, 92.1% of mites acquired M. morganii from infected bees through parasitism.

Thirdly, we investigated the transmission of M. morganii from infected Varroa mites to naïve bees. We collected the mites that had parasitized the injected bees and transferred those mites to the recipient gnotobiotic bees. We found that M. morganii can colonize and persist in most bees in the recipient group, as reflected by the CFU ($10^{5.6}$, 95% CI of $10^{4.8} \sim 10^{6.3}$, Fig 3B). Since all alive mites acquired M. morganii from the infected bees, the cumulative incidence of naïve bees acquiring M. morganii from the infected mites is 68.49%.

To infer whether the mortality of injected and parasitically transmitted M. morganii is similar, we compared the survival of bees infected through injection, to that of recipient bees. At 4 dpi, we found that the bees in the injection group showed a low survival (37.6%, 95% CI of 28.8%~49.2%), which was even lower in the recipient group (28%, 95% CI of 16.3%~48.2%). The injection group showed a high mortality as early as 2 dpi (24.7%, 95% CI of 18.1%~30.8%). Comparatively, the bees in the recipient group experienced a delay of mortality to 4 dpi, reflecting the accumulation of the bacteria. The injection may cause physical damage to the bees, reflected by the mortality in the PBS group. The bees in the PBS group showed a 76.0% survival (95% CI: 62.8%~91.9%), which is higher than the injection (Log rank test, adjusted $P<0.001$) and recipient groups (Log rank test, adjusted $P<0.001$, Fig 3C).

## Discussion

In this study, we isolated the first M. morganii in Varroa mites and highlighted that this pathogenic bacterium is highly virulent in honey bees. M. morganii is an opportunistic pathogen that infects a broad range of invertebrates [27,28], and vertebrates [29,30], including humans [31]. However, the virulence of M. morganii is rarely investigated. A previous study found that $10^5$ CFU of M. morganii resulted in 100% mortality in flies, and a lower dosage of $10^2$ CFU caused 44% mortality [32]. M. morganii also reduced the flight ability and pupal weight of the flies [33]. In nematodes, $10^6$ CFU of M. morganii leads to 100% mortality of infected nematodes in 24 hours, and $10^2$ CFU leads to 20% mortality [27]. In our data, M. morganii caused over 30% mortality in honey bees at a low dosage of $10^2$ CFU. However, the titer of M. morganii increases to $10^{10}$ CFU within 24 hours, thereby enhancing the case fatality rate, leading to the observed acute mortality.

The transmission route could determine the outcomes of the pathogen infection. For example, another opportunistic pathogen of honey bees, Serratia marcescens, caused approximately 20% mortality when infected through feeding, while injecting it leads to 100% mortality [34]. In our study, M. morganii is transmitted through the feeding of Varroa mites. At the individual level, the salivary secretions of Varroa mites can prevent the healing of the wound, through which the mites suck the hemolymph and fat body [35]. During feeding, the Varroa mites inject several pathogens into the bees, as is found in other blood-feeding arthropods [36,37]. The hosts then serve as a reservoir for the vectors to pick up and transmit the pathogens [38]. For example, Deformed wing virus (DWV) is ubiquitously harbored and vectored by the Varroa mite [39]. The titer of DWV is relatively low in honey bees when the Varroa mites are absent, which increases in orders of magnitude when the vector arrives [12,40]. In our study, we found that M. morganii couldn't be transmitted from infected bees to naïve bees during the cohousing, while it exhibited a 92.1% transmission from infected bees to mites; this is much higher than the reported 29.3% DWV transmission from bees to mites [14]. Once the Varroa mites acquire the pathogenic bacteria, they disperse the bacteria to other bees. Thus, the population of mites determines the prevalence and titer of M. morganii.

In winter, the honey bee queen stops laying eggs, and the bees cluster to keep warm [41]. The Varroa mite can easily switch from one bee to another. Thus, viruses and bacteria are dispersed among nestmates in a short period of time. Hence, the mite population determines whether a honey bee colony can survive the winter [11]. In this study, we isolated a pathogenic bacterium M. morganii, which is vectored by mites and causes substantial mortality in honey bees. This study is limited to the germfree bees, and it is interesting to investigate the impacts of gut microbes on M. morganii for future studies.

## Supporting information

**S1 Appendix. Fig A.** Heatmap of Average Nucleotide Identity of *Morganella* strains. The one isolated from the *Varroa* mite shows an average identity of 92%. The genome assembled in this study is highlighted. **Fig B.** The transmission of *M. morganii* through social contacts. We injected the engineered *M. morganii*:pBTK520 into bees and co-housed these bees with naïve ones. The engineered *M. morganii*:pBTK520 developed high titers in the injection group, which is absent in bees in the co-housing group and PBS group. The data suggest that the symbiont can't be transmitted through social contacts. **Table A.** The bacterial species isolated from mites. All mites were rinsed to remove microbes from the body surface. The mites were then homogenized to isolate single colonies under various conditions. In total, 33 bacterial species were isolated. *Morganella* is particularly interesting because it has been reported in both bees and mites in a previous metagenome study, suggesting a mutual transmission. **Table B.** The genome assembly statistics of *M. morganii* CYJ1 genome. The genome is assembled into a single contig, with 3.8 Mbp. Out of 124 conserved single-copy genes in the bacterial kingdom, 122 were identified, and the remaining 2 were fragmented. **Table C.** Genomes of *Morganella morganii* for ANI (Average Nucleotide Identity) analysis. We randomly selected 20 complete genomes to perform ANI and AAI comparisons with the strain CYJ1.
(DOCX)

**S1 Data. Data deposit.**
(XLSX)

## Acknowledgments

We appreciate the technical support of Dr. Lizhen Zhang and the platform support of Dr. Zhijiang Zeng.

## Author contributions

**Conceptualization:** Qiang Huang.

**Data curation:** Qiang Huang.

**Formal analysis:** Qiang Huang.

**Funding acquisition:** Qiang Huang.

**Investigation:** Yijun Chen.

**Writing – original draft:** Qiang Huang.

**Writing – review & editing:** Qiang Huang.

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
