## [Decision Letter · Decision Letter 0]

30 Sep 2025

PPATHOGENS-D-25-01977

Pathogenicity and transmission of Morganella morganii in honey bees

PLOS Pathogens

Dear Dr. Huang,

Thank you for submitting your manuscript to PLOS Pathogens. After careful consideration, we feel that it has merit but does not fully meet PLOS Pathogens's publication criteria as it currently stands. Therefore, we invite you to submit a revised version of the manuscript that addresses the points raised during the review process. Major consistent points were raised by all three reviewers with the writing, particularly details of the experimental protocols, the bacterial strain origin, and citations of the literature to emphasize the importance of the work. One reviewer raised concerns regarding the use of gnotobiotic bees. While we do not expect repetition of all the work in wild bees with full microbiota, this is a flaw that should be clearly acknowledged as important for future studies.

Please submit your revised manuscript within 60 days Nov 29 2025 11:59PM. If you will need more time than this to complete your revisions, please reply to this message or contact the journal office at plospathogens@plos.org. Please include the following items when submitting your revised manuscript:

We look forward to receiving your revised manuscript.

Kind regards,

Jenifer Coburn, PhD

Academic Editor

PLOS Pathogens

Jeffrey Dvorin

Section Editor

PLOS Pathogens

 Sumita Bhaduri-McIntosh

Editor-in-Chief

PLOS Pathogens

orcid.org/0000-0003-2946-9497

 Michael Malim

Editor-in-Chief

PLOS Pathogens

orcid.org/0000-0002-7699-2064

**Journal Requirements:**

At this stage, the following Authors/Authors require contributions: Yijun Chen, and Qiang Huang. Please ensure that the full contributions of each author are acknowledged in the "Add/Edit/Remove Authors" section of our submission form.

- TM on page: 16.

5) We notice that your supplementary Figures, Tables, and information are included in the manuscript file. Please remove them and upload them with the file type 'Supporting Information'. Please ensure that each Supporting Information file has a legend listed in the manuscript after the references list.

Potential Copyright Issues:

i) Please confirm (a) that you are the photographer of 1a, and 1b, or (b) provide written permission from the photographer to publish the photo(s) under our CC BY 4.0 license.

7) We note that your Data Availability Statement is currently as follows: "All study data are included in the article and supplementary material.". Please confirm at this time whether or not your submission contains all raw data required to replicate the results of your study. Authors must share the “minimal data set” for their submission. PLOS defines the minimal data set to consist of the data required to replicate all study findings reported in the article, as well as related metadata and methods (https://journals.plos.org/plosone/s/data-availability#loc-minimal-data-set-definition).

8) Please amend your detailed Financial Disclosure statement. This is published with the article. It must therefore be completed in full sentences and contain the exact wording you wish to be published.

9) Please revise your current Competing Interest statement to the standard "The authors have declared that no competing interests exist."

**Reviewers' Comments:**

Reviewer's Responses to Questions

**Part I - Summary**

Reviewer #1: The Manuscript give useful and decisive insight into the transmission pathways of M. morganii bacteria to honey bees, highlighting the role the parasitic mite Varroa destructor as its main vector. This aim is reached with a clear and simple experimental procedure, explained with parsimony of the syntax, and presented with insightful graphics. In my opinion, it will provide an asset for managing bee mortality, and modelling host-parasite dynamics. However, the Manuscript also includes quite a few weak spots, which are commonly found throughout: these include reference sourcing (most importantly, as regarding to the bacterial strain used in the experiment), methodology explanation, and a loose use of epidemiological terminology. Nonetheless, most of these soft spots only require a minor revision: I hence support this work on its way to publication, once the main issues are addressed.

Reviewer #2: The manuscript presents a well-designed study that provides potentially important insights into honey bee health and pathogens. They describe in more detail the pathogenicity and transmission of bacteria isolated from parasitic mites that can cause colony failure. The experimental methodology is appropriate and sound, and the data appear to support the main conclusion and the study could make a valuable contribution to the field. However, the manuscript suffers from weaknesses in presentation, clarity, and data availability that should be addressed/improved before it can be considered further.

Reviewer #3: The manuscript looking at the pathogenicity and transmission of Morganella morganii in honey bees made use of bacterial cultivation methods in isolating and sequencing an isolate from Varroa destructor mites. This isolate was then used to determine in pupae and gnotobiotic adults the mortality that would be caused by injecting various concentrations of bacteria. The authors also created a GFP expressing clone in this background, which they then used for looking at acquisition of the isolate by mites that were allowed to feed on infected adults that had been injected as well as transmission from the mites to naive adults. In all experiments where adults were used, they were gnotobiotic. They do show that the bacterium cannot be transferred from bee to bee when both uninfected and infected bees are house together.

**Part II – Major Issues: Key Experiments Required for Acceptance**

Reviewer #1: The bacteria investigated is a particular strain of M. morganii that you claim to have isolated in [17] (M. morganii CIJ01), as stated in L51; nonetheless, no reference to this pathogen is provided in [17]: is this reference correct? Make sure to provide a complete explanation as to the source of the organisms you have used.

Following, I list a line-by-line revision of the main body of the text, including figures and supplementary material when appropriate.

L83: I think that the last sentence is incomplete, as I suspect that you not only calculated the number of cells in the 10-7 inoculum, but also used this titer to obtain the 10-5 and 10-3 solutions as well. If so, I’d reformulate as “To estimate the number of CFUs in each dilution, we first counted the number of cells in 1µL of 10-7 inoculum (N=11 replicates); then, we derived the titer of the 10-5 and 10-3 dilution, as a proportion of the average number of CFUs counts in the 10-7 dilution.”. Also, I think this sentence would be better placed in L77, before you describe how you used the inoculum.

L167-170: Clarify how you get to 92.1%: from Figure 3, I suspect that you divided 35 infected mites by 38 (those that were transferred on infected bees): however, none of these numbers appear in the main text. The sample size (38) and the formula should be stated in Methods, and the outcome (35 out of 38) in the Results. Furthermore: unless you acknowledge the basic reproductive number (R0) of Morganella, the use of the terminology “transmission rate” is improper (as the transmission rate is R0 over time): I suggest you to refer to “cumulative incidence over X days/hours” or “attack rate over X days/hours” (See caption of Figure 3, too)

Reviewer #2: While the experimental work is strong, the manuscript suffers from weak writing and poor structure. The narrative would benefit from clearer organization, more precise framing of the research questions, and stronger transitions between sections.

The most important is that the manuscript lacks a Data Availability Statement. There also seems to be no raw data sets, which nowadays is a standard. If I somehow missed this, then the authors and the editors should ignore this comment.

The introduction is very brief and does not adequately place the work in the context of existing literature. I am missing clear questions and expectations. It should be expanded, putting the question in a broader frame

The manuscript cites only 37 references, which is limited for a study of this scope. The authors should broaden the reference list to include relevant recent work and ensure that the findings are well contextualized in the current literature.

The discussion is very basic and does not sufficiently explore the implications of the results, limitations of the study, or how findings relate to broader knowledge in the field. Authors should expand this section to highlight significance, compare results with similar studies, acknowledge limitations, and suggest directions for future research.

Line 64 – 16s sequencing is not clearly described what has been done. Also, where are those sequences? Are they deposited somewhere?

Line 68-67 – where is the genome annotation?

Line 119 – How was the genome assembled?

Line 182 – why are the PBS bees also dying?

Line 196 – citation?

Reviewer #3: 1. One of the drawbacks/concerns that I have with this work is that it has been conducted in adult bees without any microbiome (gnotobiotic). It has been shown over the last 10 years how important the microbiome of an arthropod (most organisms) is maintaining a sort of homeostasis and to keep possible pathogenic bacteria at a level that it does not impact the individuals health. This can take the form of antibacterial peptides that are bacteriostatic or other PAMPs which work in a similar manner. There Sia whole host of work done in mosquitoes and ticks displaying this.

This would then make the set of experiments here, of displaying a pathogenic potential, very one-sided. I would strongly suggest the authors use their GFP generated isolate to conduct the experiments in regular bees from a natural/lab colony that has a regular microbiome to determine if what they suggest is valid in nature especially given their discussion section.

The authors do not at all address in their discussion this missing aspect associated with the absence of a microbiota.

2. My second concern is trying to understand how many mites were used for the acquisition studies and also how many where used for the transmission studies per bee. Also were all mites that fed on naive bees infected ? What criteria was used to determine the transmission rate if not all mites were infected at the time of feeding on naive bees?

3. Overall the conclusion is not very convincing (lines 209-213) , without such an experiment being done. Bee colonies can be kept at lower temperature and mites (infected with the GFP isolate) can be introduced along with some of the bees , from the same colony, to see if this is possible. This is only a suggestion for the future.

**Part III – Minor Issues: Editorial and Data Presentation Modifications**

Reviewer #1: The study focus on the mite Varroa destructor, but its name is only mentioned once in the manuscript body: instead, the generic “mites” is used throughout; I think it would be useful to remind it here and there, either by using its specific name, or by using “Varroa mites” instead of just “mites”. This should be done especially in Introduction and Discussion, when it is not clear whether you are talking of mites “in general”, or V. destructor. One example of this ambivalence is in L41: “The mite associated virus has been extensively investigated…”.

Throughout the Manuscript, M. morganii it is referred to either as to as “M. morganii CIJ01” or, once engineered for fluorescence, M. morganii:pBTK520. While I find correct to explicit the strain when specific results are presented, most of the instances require the binomial nomenclature only, which would improve readability (e.g. in L53: “transmission of M. morganii CIJ01 depends on mites”, where this sentence is not only applicable to this strain, but to the species altogether). Similarly, I also think that you can omit the code “pBTK520” throughout the Manuscript, after you specify the engineering of the mites in Methods: the reader knows that your mites are fluorescent, and you can still remind of it when necessary by specifying “fluorescent M. morganii”, “tagged M. morganii” or alike. This would improve readability, while losing no information and highlighting the scope of your work.

Following, I list a line-by-line revision of the main body of the text, including figures and supplementary material when appropriate.

L12: Replace with "bees body fat" or "bee’s fat body tissue". Also in L26, L36

L12-14: Unclear: what virus? Is it more than one? Is it the virome?

L17: Virulence is a measure of the harm caused by a pathogen to a host; here, you seem to refer to mortality: in this case, a more appropriate terminology would be “case fatality rate”

L19: For simplicity, I suggest to replace "...social contacts. Instead, the transmission of M. morganii depends on bees." with "...social contacts, while it can be transmitted from mites to bees, and vice versa.".

L20: The transmission rate is R0 over time, so it does not apply to this study. I therefore suggest to refer to “cumulative incidence over X days/hours” or “attack rate over X days/hours”.

L21: As the present study do not analyse M. morganii-induced mortality in the field, I strongly suggest to replace “our data partially explain” with “our data aligns with”, or alike.

L27: I think "Despite this", "However", "Nonetheless", "Yet" or similar would be more adequate than "Comparatively" here

L28: As stated, it seems that mortality is transmitted to mite feeding, which is nonsense. I suggest rewriting as "A low dosage

of M. morganii, which is transmitted through mite feeding, causes high mortality…"

L29: M. morganii was well known before of this study: I strongly suggest to replace "reveal a novel" with "deepens the knowledge about a", unravel the transmission dynamics of a", or alike

L38: I think this sentence could benefit from some rearrangement: as it is, it sounds like all mites transmit DWV and "a bunch of" other viruses, sounding somewhat casual. I suggest something like "Upon feeding, the mites can inject a variety of viruses into the haemolymph of bees: among them, the deformed wing virus (DWV) is one of the most occurring, causing developmental disorders in infected individuals."

L38: As you are talking of DWV only, and refer to a publication which has studied it in V. destructor only, rewrite as "DWV has been extensively investigated because of its ubiquity in Varroa destructor".

L42: I suggest rephrasing as "...bee, as validated by fluorescent tagging, and infected individuals serve as…"

L44: As this two sentences are directly linked, I suggest not to separate them with a full stop; instead, I suggest stating it as "...bacteria: a previous 16S…"

L45: See comment in L38

L46: To avoid repetition, I suggest replacing "bees and mites" with "them"

L46-48: This paragraph needs rephrasing. These are the main issues I find in it:

1. It is not clear whether these bacteria were never cultured previously by anyone, or if it was a limitation of [15] only.

2. 3 bacterial genera are specified (amongst which the subject of the Manuscript), and “a few others”: which bacteria are you referring to? Whose transmission route and virulence remain unclear?

3. It is not clear how the fact that “there isn’t a ubiquitous bacterium harboured by each individual mite” (which mite?) links with a lesser attention to mite-vectored bacteria.

Maybe, this paragraph could be replaced by something in the lines of “Given the difficulties in cultivating these pathogens, and the complexity of the bacteria community found in Varroa mites, this field of study remains largely unexplored.”? Also, see comment for L17

L49-50: This paragraph needs rephrasing. These are the main issues I find in it:

1. As stated, it seems that E. fecalis nowadays in not isolated from dead bees, while it is.

2. As stated, it seems that isolating E. fecalis caused mortality in (already dead) bees, which is unlikely.

3. It is unclear how the last sentence follows from these premises

L52: If M. morganii CYJ01 is the strain that you have used, you cannot state that you isolated “a highly virulent strain of M. morganii CYJ01”. Maybe replace with “...a highly virulent strain of M. morganii (M. morganii CYJ01) in mites.”?

L55: Given that you are commenting your results, move to “Discussion”. As the pathogen (M. morganii) was already known, “revealed” sounds off; I’d replace with something like “Our data revealed a high lethality induced by M. morganii through mite bite: this might partially explain bee colony losses during late autumn and winter, when intraspecific contact among bees is higher.”.

L61: Explain the acronym PBS (I suppose Phosphate Buffered Saline)

L63: Which broth have you incubated “on regular incubators”, and which in CO2?

L65: Explain the acronym OD (I suppose Optical Density)

L66: Unclear: 12 replicates of which sample/passage?

L68: Specify “The DNA of a single colony of M. morganii...”

L68: Explain the acronym SDS and provide reference

L68: Explain the means of shearing DNA and provide reference of used instrumentation

L69: Provide reference for PacBio instrumentation

L69: Explain the acronym HIFI; if it is “high fidelity”, I suggest to just write “high fidelity”

L69: Provide reference for Flye assembler v.0.3-3

L69: Provide reference for NCBI PGAP

L72: Provide reference for BUSCO Database

L74: see comment for L17

L76: When you write “adjusted”, what do you mean? What was used for dilution and/or concentration of the inoculum?

L76: Briefly explain what OD600=1 means: e.g., “...we adjusted the inoculum so that its optical density would be 1 at 600 nm (OD600=1)”.

L76: What did you use for 10-3, 10-5 and 10-7 dilutions? I suspect it is PBS, but it must be stated.

L79: Specify “...to count bacterial CFU…”. Also in L83. Specify the method you have used to count CFUs (did you cultivate a known concentration of the sample? Did you manually count the colonies after 7 days?)

L81: Specify “...into gnotobiotic adult honey bees”

L86: Move the [20] reference; also, as the Supplementary material only provide details regarding the BTK procedures, move here the sentence in L95-96: “...using the Bee Microbiome Toolkit [20]. Detailed procedures are provided in Supplementary Material.”

L88: I suggest reformulating as “...can be transmitted among bees though co-housing them”.

L91: Later on, it becomes clear that the “no treatment” group is the same size of the “treatment” group; nonetheless, you should make it clear here: I’d rephrase as “The second group (15 gnotobiotic bees with 5 replicates = 75 gnotobiotic bees).

L90-94: In my opinion, this description would improve by focusing on your sample unit, and specifying the number of replicates (N=5) only once, at the end. I also suggest to rephrase for better clarity, as something like: “To this aim, we inoculated 1 µL of 10-7 M. morganii into 15 gnotobiotic bees, marking them with a colour patch placed on their thorax for identification. We co-housed these infected bees with 15 naive bees in a cup (1:1 ratio), providing five replication of the experiment (tot. 75 infected bees and 75 naive bees). As a control group, we inoculated 75 bees with 1µL PBS.

L98: Replace “vectored mites” with “infected mites”

L99-104: Here, you made a subsample of your population: you should specify how you did it (I assume it was by random selection). I also think that the overall description needs adjustments for better clarity. Here’s my suggestion: “First, we injected 25 gnotobiotic bees with the engineered M. morganii, transferring one mite to each bee’s body surface for 24 hours, providing 5 replicates of this experiment (N=125 infected bees and 125 mites): we’ll refer to this as the “injection” group. After 24 hours, we randomly selected 15 of those mites, transferring each one to a naïve bee’s body surface for 3 days (5 replicates, N=75 naive bees and 75 infected mites) as the “recipient” group. Furthermore, we injected PBS into 15 gnotobiotic bees and transferred mites to the bees’ body surface at a 1:1 ratio (5 replicates, N=75 gnotobiotic bees and 75 infected mites) as the “control” group.

L107: Provide version of R and reference

L107: Provide reference for each test, and move the [21] reference just after “...in the survival package”

L109: Provide explanation of FDR acronym, and reference

L110: Provide reference for Wilcoxon test

L114-117: Remove “Discussion” material and rephrase as something like: “We isolated 33 bacterial species (Table S1) and successfully cultivated M. morganii in LB, CBA and BHI in both regular and CO2 incubators.”

L122: Reference both Table S3 and Figure S1 (Figure S1 alone does not provide information regarding host). In Figure S2, highlight your sequence (either graphically or in the caption, e.g. by specifying “bottom Y axis”)

L123-124: Move the last sentence to “Discussion” (providing adequate references)

L125: As this is a “Results” sub-session, I’d stick to non-conclusive headings, e.g. “M. morganii mortality in pupae and adult bees”.

L126: see comment for L17

L128-133: For clarity, I’d rephrase as “...~99,5%), while the injection of M. morganii resulted in a significantly high mortality at all concentrations. Specifically, the pupae survival was the lowest in the 10-3 group (34,6%, 95% CI: 25,6%~46,8, Log-rank test p<0,001)…”.

L139-144: As all the bees are necessarily dead after homogenation, replace “dead” with “infected”, and “live” with “non-infected”. Also in the caption for Figure 1.

L143: Replace “approximate” with “similar”

L144-145: Move in “Methods” (from “To investigate…” to “…in vitro”). Furthermore, I think that “symbiont” can be quite a controversial terminology, when referring to Morganella: either provide scientific support for the symbiosis, or replace with “bacterial proliferation” (See caption of Figure1 too).

L148: v. L125

L149-151: Move to Methods

L152-154: Move to Methods

L157: Add “At the end of the experiment, M. morganii titer…”. Also, remove “of” and open parenthesis before “1012.6 cells”

L164: Move “after introducing the mites to infected bees” after “Comparatively,”

L168: Rewrite as “…M. morganii, while three dead mites were uninfected”.

L172: Rewrite as “…the mites that have parasitized the injected bees…”

L174: Add the numeric results of the CFUs in the recipient group. Also, see L167-170

L176-177: I suggest to state this affirmatively, as “To infer whether the mortality of injected and parasitically transmitted M. morganii is similar, we compared the survival of bees infected through injection, to that of recipient bees.”

L178: I suggest rewriting as “At 4 dpi, we found that bees in the injection group showed a low survival (37.6%, 95%CI: 28.8-49.2%), which was even lower in the recipient group…”.

L180: Add the numeric mortality ad 2 days after “high” (~25%).

L185: I suggest rewriting as “in mites, and highlighted that this pathogenic…”

L193: see comment for L17

L194: I suggest rewriting as “For example, another opportunistic pathogen of honeybees, Serratia marcescens, showed…”

L195: see comment for L17

L196: I suggest rewriting as “…mortality when infected through feeding, while injecting it led to 100% mortality [30].”

L197: I suggest replacing “data” with “study” or alike

L201: I suggest rewriting as “…Deformed wing virus (DNW) is ubiquitously…”. Also, specify what you mean when you say that

DWV titer is “low” when mites (or is it Varroa?) are absent (e.g. “relatively low in a given bee population”).

L205: I suggest rewriting as “…cohousing, while it exhibited a 92.1%...”. “…mites; this is much higher than the reported 29.3%...”

L212: See comment for L29

Bibliography: DOI missing for [6], [10], [12]

Reviewer #2: (No Response)

Reviewer #3: 1. There is no mention what was the colony of bees that was used to obtain the mites . If it was in nature , at least the area needs to be mentioned.

2. There is no citation for the DNA extraction method from a single colony.

3. Where were on the body of the bees was the inoculum injected?

4. Please note that for arthropod vectors they transmit a pathogen to a host but they acquire a pathogen when feeding on an infected host. Please make changes in the entire manuscript to reflect this where necessary (eg. line 98)

5. Line 155 and Line 176 grammatically incorrect, change.

6. Figure S1 - Legend - A new strain versus species was identified. Cannot be a new species if you refer to it as M. morganii.

PLOS authors have the option to publish the peer review history of their article (what does this mean? ). If published, this will include your full peer review and any attached files.

**Do you want your identity to be public for this peer review?** For information about this choice, including consent withdrawal, please see our Privacy Policy .

Reviewer #1: No

Reviewer #2: No

Reviewer #3: No

**Figure resubmission:**

 While revising your submission, we strongly recommend that you use PLOS’s NAAS tool (https://ngplosjournals.pagemajik.ai/artanalysis) to test your figure files. NAAS can convert your figure files to the TIFF file type and meet basic requirements (such as print size, resolution), or provide you with a report on issues that do not meet our requirements and that NAAS cannot fix. After uploading your figures to PLOS’s NAAS tool - https://ngplosjournals.pagemajik.ai/artanalysis, NAAS will process the files provided and display the results in the "Uploaded Files" section of the page as the processing is complete. If the uploaded figures meet our requirements (or NAAS is able to fix the files to meet our requirements), the figure will be marked as "fixed" above. If NAAS is unable to fix the files, a red "failed" label will appear above. When NAAS has confirmed that the figure files meet our requirements, please download the file via the download option, and include these NAAS processed figure files when submitting your revised manuscript.
---

## [Editor Report · Decision Letter 1]

9 Oct 2025

Dear Dr. Huang,

We are pleased to inform you that your manuscript 'Pathogenicity and transmission of Morganella morganii in honey bees' has been provisionally accepted for publication in PLOS Pathogens.

Best regards,

Jenifer Coburn, PhD

Academic Editor

PLOS Pathogens

Jeffrey Dvorin

Section Editor

PLOS Pathogens

Sumita Bhaduri-McIntosh

Editor-in-Chief

PLOS Pathogens

orcid.org/0000-0003-2946-9497

Michael Malim

Editor-in-Chief

PLOS Pathogens

orcid.org/0000-0002-7699-2064
---

## [Editor Report · Acceptance letter]

Dear Dr. Huang,

We are delighted to inform you that your manuscript, "Pathogenicity and transmission of Morganella morganii in honey bees," has been formally accepted for publication in PLOS Pathogens.

Best regards,

Sumita Bhaduri-McIntosh

Editor-in-Chief

PLOS Pathogens

orcid.org/0000-0003-2946-9497

Michael Malim

Editor-in-Chief

PLOS Pathogens

orcid.org/0000-0002-7699-2064